# Interpolated Policy Gradient: Merging On-Policy and Off-Policy Gradient Estimation for Deep Reinforcement Learning

**Shixiang Gu**
University of Cambridge
Max Planck Institute
sg717@cam.ac.uk

**Timothy Lillicrap**
DeepMind
countzero@google.com

**Zoubin Ghahramani**
University of Cambridge
Uber AI Labs
zoubin@eng.cam.ac.uk

**Richard E. Turner**
University of Cambridge
ret26@cam.ac.uk

**Bernhard Schölkopf**
Max Planck Institute
bs@tuebingen.mpg.de

**Sergey Levine**
UC Berkeley
svlevine@eecs.berkeley.edu

## Abstract

Off-policy model-free deep reinforcement learning methods using previously collected data can improve sample efficiency over on-policy policy gradient techniques. On the other hand, on-policy algorithms are often more stable and easier to use. This paper examines, both theoretically and empirically, approaches to merging on- and off-policy updates for deep reinforcement learning. Theoretical results show that off-policy updates with a value function estimator can be interpolated with on-policy policy gradient updates whilst still satisfying performance bounds. Our analysis uses control variate methods to produce a family of policy gradient algorithms, with several recently proposed algorithms being special cases of this family. We then provide an empirical comparison of these techniques with the remaining algorithmic details fixed, and show how different mixing of off-policy gradient estimates with on-policy samples contribute to improvements in empirical performance. The final algorithm provides a generalization and unification of existing deep policy gradient techniques, has theoretical guarantees on the bias introduced by off-policy updates, and improves on the state-of-the-art model-free deep RL methods on a number of OpenAI Gym continuous control benchmarks.

## 1 Introduction

Reinforcement learning (RL) studies how an agent that interacts sequentially with an environment can learn from rewards to improve its behavior and optimize long-term returns. Recent research has demonstrated that deep networks can be successfully combined with RL techniques to solve difficult control problems. Some of these include robotic control (Schulman et al., 2016; Lillicrap et al., 2016; Levine et al., 2016), computer games (Mnih et al., 2015), and board games (Silver et al., 2016). One of the simplest ways to learn a neural network policy is to collect a batch of behavior wherein the policy is used to act in the world, and then compute and apply a policy gradient update from this data. This is referred to as on-policy learning because all of the updates are made using data that was collected from the trajectory distribution induced by the current policy of the agent. It is straightforward to compute unbiased on-policy gradients, and practical on-policy gradient algorithms tend to be stable and relatively easy to use. A major drawback of such methods is that they tend to be data inefficient, because they only look at each data point once. Off-policy algorithms based on Q-learning and actor-critic learning (Sutton et al., 1999) have also proven to be an effective approach

to deep RL such as in (Mnih et al., 2015) and (Lillicrap et al., 2016). Such methods reuse samples by storing them in a memory replay buffer and train a value function or Q-function with off-policy updates. This improves data efficiency, but often at a cost in stability and ease of use.

Both on- and off-policy learning techniques have their own advantages. Most recent research has worked with on-policy algorithms *or* off-policy algorithms, and a few recent methods have sought to make use of both on- *and* off-policy data for learning (Gu et al., 2017; Wang et al., 2017; O'Donoghue et al., 2017). Such algorithms hope to gain advantages from both modes of learning, whilst avoiding their limitations. Broadly speaking, there have been two basic approaches in recently proposed algorithms that make use of both on- and off-policy data and updates. The first approach is to mix some ratio of on- and off-policy gradients or update steps in order to update a policy, as in the ACER and PGQ algorithms (Wang et al., 2017; O'Donoghue et al., 2017). In this case, there are no theoretical bounds on the error induced by incorporating off-policy updates. In the second approach, an off-policy Q critic is trained but is used as a control variate to reduce on-policy gradient variance, as in the Q-prop algorithm (Gu et al., 2017). This case does not introduce additional bias to the gradient estimator, but the policy updates do not use off-policy data.

We seek to unify these two approaches using the method of control variates. We introduce a parameterized family of policy gradient methods that interpolate between on-policy and off-policy learning. Such methods are in general biased, but we show that the bias can be bounded.We show that a number of recent methods (Gu et al., 2017; Wang et al., 2017; O'Donoghue et al., 2017) can be viewed as special cases of this more general family. Furthermore, our empirical results show that in most cases, a mix of policy gradient and actor-critic updates achieves the best results, demonstrating the value of considering interpolated policy gradients.

## 2   Preliminaries

A key component of our interpolated policy gradient method is the use of control variates to mix likelihood ratio gradients with deterministic gradient estimates obtained explicitly from a state-action critic. In this section, we summarize both likelihood ratio and deterministic gradient methods, as well as how control variates can be used to combine these two approaches.

### 2.1   On-Policy Likelihood Ratio Policy Gradient

At time $t$, the RL agent in state $s_t$ takes action $a_t$ according to its policy $\pi(a_t|s_t)$, the state transitions to $s_{t+1}$, and the agent receives a reward $r(s_t, a_t)$. For a parametrized policy $\pi_\theta$, the objective is to maximize the $\gamma$-discounted cumulative future return $J(\theta) = J(\pi) = \mathbb{E}_{s_0, a_0, \cdots \sim \pi}[\sum_{t=0}^{\infty} \gamma^t r(s_t, a_t)]$. Monte Carlo policy gradient methods, such as REINFORCE (Williams, 1992) and TRPO (Schulman et al., 2015), use the *likelihood ratio policy gradient* of the RL objective,

$$\nabla_\theta J(\theta) = \mathbb{E}_{\rho^\pi, \pi}[\nabla_\theta \log \pi_\theta(a_t|s_t)(\hat{Q}(s_t, a_t) - b(s_t))] = \mathbb{E}_{\rho^\pi, \pi}[\nabla_\theta \log \pi_\theta(a_t|s_t)\hat{A}(s_t, a_t)], \quad (1)$$

where $\hat{Q}(s_t, a_t) = \sum_{t'=t}^{\infty} \gamma^{t'-t} r(s_{t'}, a_{t'})$ is the Monte Carlo estimate of the "critic" $Q^\pi(s_t, a_t) = \mathbb{E}_{s_{t+1}, a_{t+1}, \cdots \sim \pi|s_t, a_t}[\hat{Q}(s_t, a_t)]$, and $\rho^\pi = \sum_{t=0}^{\infty} \gamma^t p(s_t = s)$ are the unnormalized state visitation frequencies, while $b(s_t)$ is known as the baseline, and serves to reduce the variance of the gradient estimate (Williams, 1992). If the baseline estimates the value function, $V^\pi(s_t) = \mathbb{E}_{a_t \sim \pi(\cdot|s_t)}[Q^\pi(s_t, a_t)]$, then $\hat{A}(s_t)$ is an estimate of the advantage function $A^\pi(s_t, a_t) = Q^\pi(s_t, a_t) - V^\pi(s_t)$. Likelihood ratio policy gradient methods use unbiased gradient estimates (except for the technicality detailed by Thomas (2014)), but they often suffer from high variance and are sample-intensive.

### 2.2   Off-Policy Deterministic Policy Gradient

Policy gradient methods with function approximation (Sutton et al., 1999), or actor-critic methods, are a family of policy gradient methods which first estimate the critic, or the value, of the policy by $Q_w \approx Q^\pi$, and then greedily optimize the policy $\pi_\theta$ with respect to $Q_w$. While it is not necessary for such algorithms to be off-policy, we primarily analyze the off-policy variants, such as (Riedmiller, 2005; Degris et al., 2012; Heess et al., 2015; Lillicrap et al., 2016). For example, DDPG Lillicrap et al. (2016), which optimizes a continuous deterministic policy $\pi_\theta(a_t|s_t) = \delta(a_t = \mu_\theta(s_t))$, can be summarized by the following update equations, where $Q'$ denotes the target Q network and $\beta$ denotes

| $\beta$ | $\nu$ | CV | Examples |
|---|---|---|---|
| - | 0 | No | REINFORCE (Williams, 1992),TRPO (Schulman et al., 2015) |
| $\pi$ | 0 | Yes | Q-Prop (Gu et al., 2017) |
| - | 1 | - | DDPG (Silver et al., 2014; Lillicrap et al., 2016),SVG(0) (Heess et al., 2015) |
| $\neq \pi$ | - | No | $\approx$PGQ (O'Donoghue et al., 2017), $\approx$ACER (Wang et al., 2017) |

Table 1: Prior policy gradient method objectives as special cases of IPG.

some off-policy distribution, e.g. from experience replay (Lillicrap et al., 2016):

$$w \leftarrow \arg\min \mathbb{E}_\beta[(Q_w(s_t, a_t) - y_t)^2], \quad y_t = r(s_t, a_t) + \gamma Q'(s_{t+1}, \mu_\theta(s_{t+1}))$$
$$\theta \leftarrow \arg\max \mathbb{E}_\beta[Q_w(s_t, \mu_\theta(s_t))]. \tag{2}$$

This provides the following *deterministic policy gradient* through the critic:

$$\nabla_\theta J(\theta) \approx \mathbb{E}_{\rho^\beta}[\nabla_\theta Q_w(s_t, \mu_\theta(s_t))]. \tag{3}$$

This policy gradient is generally biased due to the imperfect estimator $Q_w$ and off-policy state sampling from $\beta$. Off-policy actor-critic algorithms therefore allow training the policy on off-policy samples, at the cost of introducing potentially unbounded bias into the gradient estimate. This usually makes off-policy algorithms less stable during learning, compared to on-policy algorithms using a large batch size for each update (Duan et al., 2016; Gu et al., 2017).

## 2.3 Off-Policy Control Variate Fitting

The control variates method (Ross, 2006) is a general technique for variance reduction of a Monte Carlo estimator by exploiting a correlated variable for which we know more information such as analytical expectation. General control variates for RL include state-action baselines, and an example can be an off-policy fitted critic $Q_w$. Q-Prop (Gu et al., 2017), for example, used $\tilde{Q}_w$, the first-order Taylor expansion of $Q_w$, as the control variates, and showed improvement in stability and sample efficiency of policy gradient methods. $\mu_\theta$ here corresponds to the mean of the stochastic policy $\pi_\theta$.

$$\nabla_\theta J(\theta) = \mathbb{E}_{\rho^\pi, \pi}[\nabla_\theta \log \pi_\theta(a_t|s_t)(\hat{Q}(s_t, a_t) - \tilde{Q}_w(s_t, a_t))] + \mathbb{E}_{\rho^\pi}[\nabla_\theta Q_w(s_t, \mu_\theta(s_t))]. \tag{4}$$

The gradient estimator combines both likelihood ratio and deterministic policy gradients in Eq. 1 and 3. It has lower variance and stable gradient estimates and enables more sample-efficient learning. However, one limitation of Q-Prop is that it uses only on-policy samples for estimating the policy gradient. This ensures that the Q-Prop estimator remains unbiased, but limits the use of off-policy samples for further variance reduction.

## 3 Interpolated Policy Gradient

Our proposed approach, interpolated policy gradient (IPG), mixes likelihood ratio gradient with $\hat{Q}$, which provides unbiased but high-variance gradient estimation, and deterministic gradient through an off-policy fitted critic $Q_w$, which provides low-variance but biased gradients. IPG directly interpolates the two terms from Eq. 1 and 3:

$$\nabla_\theta J(\theta) \approx (1 - \nu)\mathbb{E}_{\rho^\pi, \pi}[\nabla_\theta \log \pi_\theta(a_t|s_t)\hat{A}(s_t, a_t)] + \nu\mathbb{E}_{\rho^\beta}[\nabla_\theta \bar{Q}_w^\pi(s_t)], \tag{5}$$

where we generalized the deterministic policy gradient through the critic as $\nabla_\theta \bar{Q}_w(s_t) = \nabla_\theta \mathbb{E}_\pi[Q_w^\pi(s_t, \cdot)]$. This generalization is to make our analysis applicable with more general forms of the critic-based control variates, as discussed in the Appendix. This gradient estimator is biased from two sources: off-policy state sampling $\rho^\beta$, and inaccuracies in the critic $Q_w$. However, as we show in Section 4, we can bound the biases for all the cases, and in some cases, the algorithm still guarantees monotonic convergence as in Kakade & Langford (2002); Schulman et al. (2015).

### 3.1 Control Variates for Interpolated Policy Gradient

While IPG includes $\nu$ to trade off bias and variance directly, it contains a likelihood ratio gradient term, for which we can introduce a control variate (CV) Ross (2006) to further reduce the estimator variance.

The expression for the IPG with control variates is below, where $A_w^\pi(s_t, a_t) = Q_w(s_t, a_t) - \bar{Q}_w^\pi(s_t)$,

$$
\begin{aligned}
\nabla_\theta J(\theta) &\approx (1-\nu)\mathbb{E}_{\rho^\pi,\pi}[\nabla_\theta \log \pi_\theta(a_t|s_t)\hat{A}(s_t, a_t)] + \nu\mathbb{E}_{\rho^\beta}[\nabla_\theta \bar{Q}_w^\pi(s_t)] \\
&= (1-\nu)\mathbb{E}_{\rho^\pi,\pi}[\nabla_\theta \log \pi_\theta(a_t|s_t)(\hat{A}(s_t, a_t) - A_w^\pi(s_t, a_t))] \\
&\quad + (1-\nu)\mathbb{E}_{\rho^\pi}[\nabla_\theta \bar{Q}_w^\pi(s_t)] + \nu\mathbb{E}_{\rho^\beta}[\nabla_\theta \bar{Q}_w^\pi(s_t)] \\
&\approx (1-\nu)\mathbb{E}_{\rho^\pi,\pi}[\nabla_\theta \log \pi_\theta(a_t|s_t)(\hat{A}(s_t, a_t) - A_w^\pi(s_t, a_t))] + \mathbb{E}_{\rho^\beta}[\nabla_\theta \bar{Q}_w^\pi(s_t)].
\end{aligned}
\tag{6}
$$

The first approximation indicates the biased approximation from IPG, while the second approximation indicates replacing the $\rho^\pi$ in the control variate correction term with $\rho^\beta$ and merging with the last term. The second approximation is a design decision and introduces additional bias when $\beta \neq \pi$ but it helps simplify the expression to be analyzed more easily, and the additional benefit from the variance reduction from the control variate could still outweigh this extra bias. The biases are analyzed in Section 4. The likelihood ratio gradient term is now proportional to the residual in on- and off-policy advantage estimates $\hat{A}(s_t, a_t) - A_w^\pi(s_t, a_t)$, and therefore, we call this term *residual likelihood ratio gradient*. Intuitively, if the off-policy critic estimate is accurate, this term has a low magnitude and the overall variance of the estimator is reduced.

### 3.2 Relationship to Prior Policy Gradient and Actor-Critic Methods

Crucially, IPG allows interpolating a rich list of prior deep policy gradient methods using only three parameters: $\beta$, $\nu$, and the use of the control variate (CV). The connection is summarized in Table 1 and the algorithm is presented in Algorithm 1. Importantly, a wide range of prior work has only explored limiting cases of the spectrum, e.g. $\nu = 0, 1$, with or without the control variate. Our work provides a thorough theoretical analysis of the biases, and in some cases performance guarantees, for each of the method in this spectrum and empirically demonstrates often the best performing algorithms are in the midst of the spectrum.

---

**Algorithm 1** Interpolated Policy Gradient

**input** $\beta$, $\nu$, useCV
1: Initialize $w$ for critic $Q_w$, $\theta$ for stochastic policy $\pi_\theta$, and replay buffer $\mathcal{R} \leftarrow \emptyset$.
2: **repeat**
3:     Roll-out $\pi_\theta$ for $E$ episodes, $T$ time steps each, to collect a batch of data $\mathcal{B} = \{s, a, r\}_{1:T,1:E}$ to $\mathcal{R}$
4:     Fit $Q_w$ using $\mathcal{R}$ and $\pi_\theta$, and fit baseline $V_\phi(s_t)$ using $\mathcal{B}$
5:     Compute Monte Carlo advantage estimate $\hat{A}_{t,e}$ using $\mathcal{B}$ and $V_\phi$
6:     **if** useCV **then**
7:         Compute critic-based advantage estimate $\bar{A}_{t,e}$ using $\mathcal{B}$, $Q_w$ and $\pi_\theta$
8:         Compute and center the learning signals $l_{t,e} = \hat{A}_{t,e} - \bar{A}_{t,e}$ and set $b = 1$
9:     **else**
10:        Center the learning signals $l_{t,e} = \hat{A}_{t,e}$ and set $b = \nu$
11:     **end if**
12:    Multiply $l_{t,e}$ by $(1-\nu)$
13:    Sample $\mathcal{D} = s_{1:M}$ from $\mathcal{R}$ and/or $\mathcal{B}$ based on $\beta$
14:    Compute $\nabla_\theta J(\theta) \approx \frac{1}{ET}\sum_e\sum_t \nabla_\theta \log \pi_\theta(a_{t,e}|s_{t,e})l_{t,e} + \frac{b}{M}\sum_m \nabla_\theta \bar{Q}_w^\pi(s_m)$
15:    Update policy $\pi_\theta$ using $\nabla_\theta J(\theta)$
16: **until** $\pi_\theta$ converges.

---

### 3.3 $\nu = 1$: Actor-Critic methods

Before presenting our theoretical analysis, an important special case to discuss is $\nu = 1$, which corresponds to a deterministic actor-critic method. Several advantages of this special case include that the policy can be deterministic and the learning can be done completely off-policy, as it does not have to estimate the on-policy Monte Carlo critic $\hat{Q}$. Prior work such as DDPG Lillicrap et al. (2016) and related Q-learning methods have proposed aggressive off-policy exploration strategy to exploit these properties of the algorithm. In this work, we compare alternatives such as using on-policy exploration and stochastic policy with classical DDPG algorithm designs, and show that in some domains the off-policy exploration can significantly deteriorate the performance. Theoretically, we confirm this empirical observation by showing that the bias from off-policy sampling in $\beta$ increases

monotonically with the total variation or KL divergence between $\beta$ and $\pi$. Both the empirical and theoretical results indicate that well-designed actor-critic methods with an on-policy exploration strategy could be a more reliable alternative than with an on-policy exploration.

## 4 Theoretical Analysis

In this section, we present a theoretical analysis of the bias in the interpolated policy gradient. This is crucial, since understanding the biases of the methods can improve our intuition about its performance and make it easier to design new algorithms in the future. Because IPG includes many prior methods as special cases, our analysis also applies to those methods and other intermediate cases. We first analyze a special case and derive results for general IPG. All proofs are in the Appendix.

### 4.1 $\beta \neq \pi$, $\nu = 0$: Policy Gradient with Control Variate and Off-Policy Sampling

This section provides an analysis of the special case of IPG with $\beta \neq \pi$, $\nu = 1$, and the control variate. Plugging in to Eq. 6, we get an expression similar to Q-Prop in Eq. 4,

$$\nabla_\theta J(\theta) \approx \mathbb{E}_{\rho^\pi, \pi}[\nabla_\theta \log \pi_\theta(a_t|s_t)(\hat{A}(s_t, a_t) - A_w^\pi(s_t, a_t))] + \mathbb{E}_{\rho^\beta}[\nabla_\theta \bar{Q}_w^\pi(s_t)], \qquad (7)$$

except that it also supports utilizing off-policy data for updating the policy. To analyze the bias for this gradient expression, we first introduce $\tilde{J}(\pi, \tilde{\pi})$, a local approximation to $J(\pi)$, which has been used in prior theoretical work (Kakade & Langford, 2002; Schulman et al., 2015). The derivation and the bias from this approximation are discussed in the proof for Theorem 1 in the Appendix.

$$J(\pi) = J(\tilde{\pi}) + \mathbb{E}_{\rho^\pi, \pi}[A^{\tilde{\pi}}(s_t, a_t)] \approx J(\tilde{\pi}) + \mathbb{E}_{\rho^{\tilde{\pi}}, \pi}[A^{\tilde{\pi}}(s_t, a_t)] = \tilde{J}(\pi, \tilde{\pi}). \qquad (8)$$

Note that $J(\pi) = \tilde{J}(\pi, \tilde{\pi} = \pi)$ and $\nabla_\pi J(\pi) = \nabla_\pi \tilde{J}(\pi, \tilde{\pi} = \pi)$. In practice, $\tilde{\pi}$ corresponds to policy $\pi_k$ at iteration $k$ and $\pi$ corresponds next policy $\pi_{k+1}$ after parameter update. Thus, this approximation is often sufficiently good. Next, we write the approximate objective for Eq. 7,

$$\tilde{J}^{\beta, \nu=0, CV}(\pi, \tilde{\pi}) \triangleq J(\tilde{\pi}) + \mathbb{E}_{\rho^{\tilde{\pi}}, \pi}[A^{\tilde{\pi}}(s_t, a_t) - A_w^{\tilde{\pi}}(s_t, a_t)] + \mathbb{E}_{\rho^\beta}[\bar{A}_w^{\pi, \tilde{\pi}}(s_t)] \approx \tilde{J}(\pi, \tilde{\pi})$$
$$\bar{A}_w^{\pi, \tilde{\pi}}(s_t) = \mathbb{E}_\pi[A_w^{\tilde{\pi}}(s_t, \cdot)] = \mathbb{E}_\pi[Q_w(s_t, \cdot)] - \mathbb{E}_{\tilde{\pi}}[Q_w(s_t, \cdot)]. \qquad (9)$$

Note that $\tilde{J}^{\beta, \nu=0}(\pi, \tilde{\pi} = \pi) = \tilde{J}(\pi, \tilde{\pi} = \pi) = J(\pi)$, and $\nabla_\pi \tilde{J}^{\beta, \nu=0}(\pi, \tilde{\pi} = \pi)$ equals Eq. 7. We can bound the absolute error between $\tilde{J}^{\beta, \nu=0, CV}(\pi, \tilde{\pi})$ and $J(\pi)$ by the following theorem, where $D_{KL}^{\max}(\pi_i, \pi_j) = \max_s D_{KL}(\pi_i(\cdot|s), \pi_j(\cdot|s))$ is the maximum KL divergence between $\pi_i, \pi_j$.

**Theorem 1.** *If* $\epsilon = \max_s |\bar{A}_w^{\pi, \tilde{\pi}}(s)|, \zeta = \max_s |\bar{A}^{\pi, \tilde{\pi}}(s)|$, *then*

$$\left\| J(\pi) - \tilde{J}^{\beta, \nu=0, CV}(\pi, \tilde{\pi}) \right\|_1 \leq 2\frac{\gamma}{(1-\gamma)^2}\left(\epsilon\sqrt{D_{KL}^{\max}(\tilde{\pi}, \beta)} + \zeta\sqrt{D_{KL}^{\max}(\pi, \tilde{\pi})}\right)$$

Theorem 1 contains two terms: the second term confirms $\tilde{J}^{\beta, \nu=0, CV}$ is a local approximation around $\pi$ and deviates from $J(\pi)$ as $\tilde{\pi}$ deviates, and the first term bounds the bias from off-policy sampling using the KL divergence between the policies $\tilde{\pi}$ and $\beta$. This means that the algorithm fits well with policy gradient methods which constrain the KL divergence per policy update, such as covariant policy gradient (Bagnell & Schneider, 2003), natural policy gradient (Kakade & Langford, 2002), REPS (Peters et al., 2010), and trust-region policy optimization (TRPO) (Schulman et al., 2015).

#### 4.1.1 Monotonic Policy Improvement Guarantee

Some forms of on-policy policy gradient methods have theoretical guarantees on monotonic convergence Kakade & Langford (2002); Schulman et al. (2015). Such guarantees often correspond to stable empirical performance on challenging problems, even when some of the constraints are relaxed in practice (Schulman et al., 2015; Duan et al., 2016; Gu et al., 2017). We can show that a variant of IPG allows off-policy sampling while still guaranteeing monotonic convergence. The algorithm and the proof are provided in the appendix. This algorithm is usually impractical to implement; however, IPG with trust-region updates when $\beta \neq \pi, \nu = 1, \text{CV} = true$ approximates this monotonic algorithm, similar to how TRPO is an approximation to the theoretically monotonic algorithm proposed by Schulman et al. (2015).

## 4.2 General Bounds on the Interpolated Policy Gradient

We can establish bias bounds for the general IPG algorithm, with and without the control variate, using Theorem 2. The additional term that contributes to the bias in the general case is $\delta$, which represents the error between the advantage estimated by the off-policy critic and the true $A^\pi$ values.

**Theorem 2.** *If* $\delta = \max_{s,a} |A^{\tilde{\pi}}(s,a) - A_w^{\tilde{\pi}}(s,a)|$, $\epsilon = \max_s |\bar{A}_w^{\pi,\tilde{\pi}}(s)|$, $\zeta = \max_s |\bar{A}^{\pi,\tilde{\pi}}(s)|$,

$$\tilde{J}^{\beta,\nu}(\pi,\tilde{\pi}) \triangleq J(\tilde{\pi}) + (1-\nu)\mathbb{E}_{\rho^{\tilde{\pi}},\pi}[\hat{A}^{\tilde{\pi}}] + \nu\mathbb{E}_{\rho^\beta}[\bar{A}_w^{\pi,\tilde{\pi}}]$$

$$\tilde{J}^{\beta,\nu,CV}(\pi,\tilde{\pi}) \triangleq J(\tilde{\pi}) + (1-\nu)\mathbb{E}_{\rho^{\tilde{\pi}},\pi}[\hat{A}^{\tilde{\pi}} - A_w^{\tilde{\pi}}] + \mathbb{E}_{\rho^\beta}[\bar{A}_w^{\pi,\tilde{\pi}}]$$

*then,* $\left\| J(\pi) - \tilde{J}^{\beta,\nu}(\pi,\tilde{\pi}) \right\|_1 \leq \dfrac{\nu\delta}{1-\gamma} + 2\dfrac{\gamma}{(1-\gamma)^2}\left(\nu\epsilon\sqrt{D_{KL}^{\max}(\tilde{\pi},\beta)} + \zeta\sqrt{D_{KL}^{\max}(\pi,\tilde{\pi})}\right)$

$$\left\| J(\pi) - \tilde{J}^{\beta,\nu,CV}(\pi,\tilde{\pi}) \right\|_1 \leq \dfrac{\nu\delta}{1-\gamma} + 2\dfrac{\gamma}{(1-\gamma)^2}\left(\epsilon\sqrt{D_{KL}^{\max}(\tilde{\pi},\beta)} + \zeta\sqrt{D_{KL}^{\max}(\pi,\tilde{\pi})}\right)$$

This bound shows that the bias from directly mixing the deterministic policy gradient through $\nu$ comes from two terms: how well the critic $Q_w$ is approximating $Q^\pi$, and how close the off-policy sampling policy is to the actor policy. We also show that the bias introduced is proportional to $\nu$ while the variance of the high variance likelihood ratio gradient term is proportional to $(1-\nu)^2$, so $\nu$ allows directly trading off bias and variance. Theorem 2 fully bounds bias in the full spectrum of IPG methods; this enables us to analyze how biases arise and interact and help us design better algorithms.

## 5 Related Work

An overarching aim of this paper is to help unify on-policy and off-policy policy gradient algorithms into a single conceptual framework. Our analysis examines how Q-Prop (Gu et al., 2017), PGQ (O'Donoghue et al., 2017), and ACER (Wang et al., 2017), which are all recent works that combine on-policy with off-policy learning, are connected to each other (see Table 1). IPG with $0 < \nu < 1$ and without the control variate relates closely to PGQ and ACER, but differ in the details. PGQ mixes in the Q-learning Bellman error objective, and ACER mixes parameter update steps rather than directly mixing gradients. And both PGQ and ACER come with numerous additional design details that make fair comparisons with methods like TRPO and Q-Prop difficult. We instead focus on the three minimal variables of IPG and explore their settings in relation to the closely related TRPO and Q-Prop methods, in order to theoretically and empirically understand in which situations we might expect gains from mixing on- and off-policy gradients.

Asides from these more recent works, the use of off-policy samples with policy gradients has been a popular direction of research (Peshkin & Shelton, 2002; Jie & Abbeel, 2010; Degris et al., 2012; Levine & Koltun, 2013). Most of these methods rely on variants of importance sampling (IS) to correct for bias. The use of importance sampling ensures unbiased estimates, but at the cost of considerable variance, as quantified by the ESS measure used by Jie & Abbeel (2010). Ignoring importance weights produces bias but, as shown in our analysis, this bias can be bounded. Therefore, our IPG estimators have higher bias as the sampling distribution deviates from the policy, while IS methods have higher variance. Among these importance sampling methods, Levine & Koltun (2013) evaluates on tasks that are the most similar to our paper, but the focus is on using importance sampling to include demonstrations, rather than to speed up learning from scratch.

Lastly, there are many methods that combine on- and off-policy data for policy evaluation (Precup, 2000; Mahmood et al., 2014; Munos et al., 2016), mostly through variants of importance sampling. Combining our methods with more sophisticated policy evaluation methods will likely lead to further improvements, as done in (Degris et al., 2012). A more detailed analysis of the effect of importance sampling on bias and variance is left to future work, where some of the relevant work includes Precup (2000); Jie & Abbeel (2010); Mahmood et al. (2014); Jiang & Li (2016); Thomas & Brunskill (2016).

## 6 Experiments

In this section, we empirically show that the three parameters of IPG can interpolate different behaviors and often achieve superior performance versus prior methods that are limiting cases of this

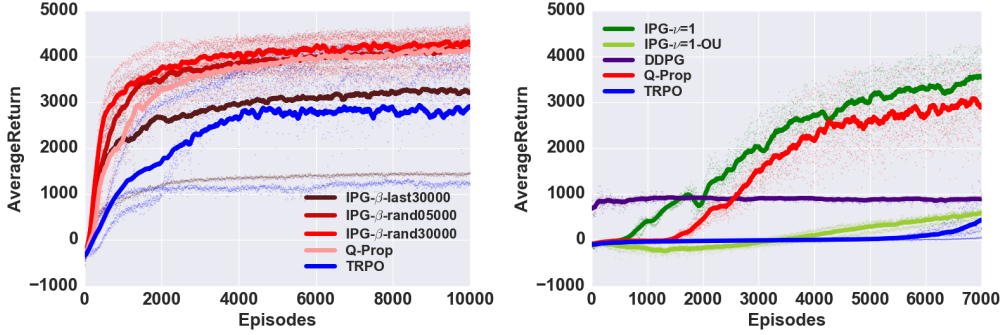

(a) IPG with $\nu = 0$ and the control variate.  (b) IPG with $\nu = 1$.

Figure 1: (a) IPG-$\nu = 0$ vs Q-Prop on HalfCheetah-v1, with batch size 5000. IPG-$\beta$-rand30000, which uses 30000 random samples from the replay as samples from $\beta$, outperforms Q-Prop in terms of learning speed. (b) IPG-$\nu$=1 vs other algorithms on Ant-v1. In this domain, on-policy IPG-$\nu$=1 with on-policy exploration significantly outperforms DDPG and IPG-$\nu$=1-OU, which use a heuristic OU (Ornstein–Uhlenbeck) process noise exploration strategy, and marginally outperforms Q-Prop.

approach. Crucially, all methods share the same algorithmic structure as Algorithm 1, and we hold the rest of the experimental details fixed. All experiments were performed on MuJoCo domains in OpenAI Gym (Todorov et al., 2012; Brockman et al., 2016), with results presented for the average over three seeds. Additional experimental details are provided in the Appendix.

## 6.1  $\beta \neq \pi$, $\nu = 0$, with the control variate

We evaluate the performance of the special case of IPG discussed in Section 4.1. This case is of particular interest, since we can derive monotonic convergence results for a variant of this method under certain conditions, despite the presence of off-policy updates. Figure 1a shows the performance on the HalfCheetah-v1 domain, when the policy update batch size is 5000 transitions (i.e. 5 episodes). "last" and "rand" indicate if $\beta$ samples from the most recent transitions or uniformly from the experience replay. "last05000" would be equivalent to Q-Prop given $\nu = 0$. Comparing "IPG-$\beta$-rand05000" and "Q-Prop" curves, we observe that by drawing the same number of samples randomly from the replay buffer for estimating the critic gradient, instead of using the on-policy samples, we get faster convergence. If we sample batches of size 30000 from the replay buffer, the performance further improves. However, as seen in the "IPG-$\beta$-last30000" curve, if we instead use the 30000 most recent samples, the performance degrades. One possible explanation for this is that, while using random samples from the replay increases the bound on the bias according to Theorem 1, it also decorrelates the samples within the batch, providing more stable gradients. This is the original motivation for experience replay in the DQN method (Mnih et al., 2015), and we have shown that such decorrelated off-policy samples can similarly produce gains for policy gradient algorithms. See Table 2 for results on other domains.

The results for this variant of IPG demonstrate that random sampling from the replay provides further improvement on top of Q-Prop. Note that these replay buffer samples are different from standard off-policy samples in DDPG or DQN algorithms, which often use aggressive heuristic exploration strategies. The samples used by IPG are sampled from prior policies that follow a conservative trust-region update, resulting in greater regularity but less exploration. In the next section, we show that in some cases, ensuring that the off-policy samples are not *too* off-policy is essential for good performance.

## 6.2  $\beta = \pi$, $\nu = 1$

In this section, we empirically evaluate another special case of IPG, where $\beta = \pi$, indicating on-policy sampling, and $\nu = 1$, which reduces to a trust-region, on-policy variant of a deterministic actor-critic method. Although this algorithm performs actor-critic updates, the use of a trust region makes it more similar to TRPO or Q-Prop than DDPG.

| | HalfCheetah-v1 | | Ant-v1 | | Walker-v1 | | Humanoid-v1 | |
|---|---|---|---|---|---|---|---|---|
| | $\beta = \pi$ | $\beta \neq \pi$ | $\beta = \pi$ | $\beta \neq \pi$ | $\beta = \pi$ | $\beta \neq \pi$ | $\beta = \pi$ | $\beta \neq \pi$ |
| IPG-$\nu$=0.2 | 3356 | 3458 | **4237** | **4415** | **3047** | 1932 | 1231 | 920 |
| IPG-cv-$\nu$=0.2 | **4216** | 4023 | **3943** | **3421** | 1896 | 1411 | **1651** | **1613** |
| IPG-$\nu$=1 | 2962 | **4767** | **3469** | **3780** | 2704 | 805 | **1571** | **1530** |
| Q-Prop | 4178 | **4182** | 3374 | **3479** | 2832 | 1692 | 1423 | **1519** |
| TRPO | 2889 | N.A. | 1520 | N.A. | 1487 | N.A. | 615 | N.A. |

Table 2: Comparisons on all domains with mini-batch size 10000 for Humanoid and 5000 otherwise. We compare the maximum of average test rewards in the first 10000 episodes (Humanoid requires more steps to fully converge; see the Appendix for learning curves). Results outperforming Q-Prop (or IPG-cv-$\nu$=0 with $\beta = \pi$) are boldface. The two columns show results with on-policy and off-policy samples for estimating the deterministic policy gradient.

Results for all domains are shown in Table 2. Figure 1b shows the learning curves on Ant-v1. Although IPG-$\nu$=1 methods can be off-policy, the policy is updated every 5000 samples to keep it consistent with other IPG methods, while DDPG updates the policy on every step in the environment and makes other design choices Lillicrap et al. (2016). We see that, in this domain, standard DDPG becomes stuck with a mean reward of 1000, while IPG-$\nu$=1 improves monotonically, achieving a significantly better result. To investigate why this large discrepancy arises, we also ran IPG-$\nu$=1 with the same OU process exploration noise as DDPG, and observed large degradation in performance. This provides empirical support for Theorem 2. It is illuminating to contrast this result with the previous experiment, where the off-policy samples did not adversely alter the results. In the previous experiments, the samples came from Gaussian policies updated with trust-regions. The difference between $\pi$ and $\beta$ was therefore approximately bounded by the trust-regions. In the experiment with Brownian noise, the behaving policy uses temporally correlated noise, with potentially unbounded KL-divergence from the learned Gaussian policy. In this case, the off-policy samples result in excessive bias, wiping out the variance reduction benefits of off-policy sampling. In general, we observed that for the harder Ant-v1 and Walker-v1 domains, on-policy exploration is more effective, even when doing off-policy state sampling from a replay buffer. This results suggests the following lesson for designing off-policy actor-critic methods: for domains where exploration is difficult, it may be more effective to use on-policy exploration with bounded policy updates than to design heuristic exploration rules such as the OU process noise, due to the resulting reduction in bias.

### 6.3 General Cases of Interpolated Policy Gradient

Table 2 shows the results for experiments where we compare IPG methods with varying values of $\nu$; additional results are provided in the Appendix. $\beta \neq \pi$ indicates that the method uses off-policy samples from the replay buffer, with the same batch size as the on-policy batch for fair comparison. We ran sweeps over $\nu = \{0.2, 0.4, 0.6, 0.8\}$ and found that $\nu = 0.2$ consistently produce better performance than Q-Prop, TRPO or prior actor-critic methods. This is consistent with the results in PGQ (O'Donoghue et al., 2017) and ACER (Wang et al., 2017), which found that their equivalent of $\nu = 0.1$ performed best on their benchmarks. Importantly, we compared all methods with the same algorithm designs (exploration, policy, etc.), since Q-Prop and TRPO are IPG-$\nu$=0 with and without the control variate. IPG-$\nu$=1 is a novel variant of the actor-critic method that differs from DDPG (Lillicrap et al., 2016) and SVG(0) (Heess et al., 2015) due to the use of a trust region. The results in Table 2 suggest that, in most cases, the best performing algorithm is one that interpolates between the policy-gradient and actor-critic variants, with intermediate values of $\nu$.

## 7 Discussion

In this paper, we introduced interpolated policy gradient methods, a family of policy gradient algorithms that allow mixing off-policy learning with on-policy learning while satisfying performance bounds. This family of algorithms unifies and interpolates on-policy likelihood ratio policy gradient and off-policy deterministic policy gradient, and includes a number of prior works as approximate limiting cases. Empirical results confirm that, in many cases, interpolated gradients have improved sample-efficiency and stability over the prior state-of-the-art methods, and the theoretical results provide intuition for analyzing the cases in which the different methods perform well or poorly. Our hope is that this detailed analysis of interpolated gradient methods can not only provide for more effective algorithms in practice, but also give useful insight for future algorithm design.

**Acknowledgements**

This work is supported by generous sponsorship from Cambridge-Tübingen PhD Fellowship, NSERC, and Google Focused Research Award.

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
