[Supplementary Material]

# Interpolated Policy Gradient: Appendix

**Shixiang Gu**
University of Cambridge
Max Planck Institute
sg717@cam.ac.uk

**Timothy Lillicrap**
DeepMind
countzero@google.com

**Zoubin Ghahramani**
University of Cambridge
Uber AI Labs
zoubin@eng.cam.ac.uk

**Richard E. Turner**
University of Cambridge
ret26@cam.ac.uk

**Bernhard Schölkopf**
Max Planck Institute
bs@tuebingen.mpg.de

**Sergey Levine**
UC Berkeley
svlevine@eecs.berkeley.edu

## A  Proof for Theorem 1

### A.1  Local approximation objective with bounded bias

In the main paper, we introduced the approximate objective $\tilde{J}(\pi, \tilde{\pi})$ to $J(\pi)$ during our theoretical analysis. In this section, we discuss the motivations behind this choice, referencing the prior work (Kakade & Langford, 2002; Schulman et al., 2015).

First, the expected return $J(\pi)$ of a policy $\pi$ can be written as the sum of the expected return $J(\tilde{\pi})$ of another policy $\tilde{\pi}$ and the expected advantage term between the two policies in the equation, where $A^{\tilde{\pi}}(s_t, a_t)$ is the advantage of policy $\tilde{\pi}$,

$$J(\pi) = J(\tilde{\pi}) + \mathbb{E}_{\rho^\pi, \pi}[A^{\tilde{\pi}}(s_t, a_t)].$$

For the proof, see Lemma 1 in (Schulman et al., 2015). This expression is still not tractable to analyze because of the dependency of unnormalized state sampling distribution $\rho^\pi$ on $\pi$. Kakade & Langford (2002); Schulman et al. (2015) thus introduce a local approximation by replacing $\rho^\pi$ with $\rho^\pi$,

$$J(\pi) \approx J(\tilde{\pi}) + \mathbb{E}_{\rho^{\tilde{\pi}}, \pi}[A^{\tilde{\pi}}(s_t, a_t)] \triangleq \tilde{J}(\pi, \tilde{\pi}).$$

We can show that $J(\pi) = \tilde{J}(\pi, \tilde{\pi} = \pi)$ and $\nabla_\pi J(\pi) = \nabla_\pi \tilde{J}(\pi, \tilde{\pi} = \pi)$, meaning that the $J(\pi)$ and $\tilde{J}(\pi, \tilde{\pi})$ match up to the first order terms. Schulman et al. (2015) then uses this property, in combination with minorization-maximization Hunter & Lange (2004), to derive a monotonic convergence proof for a variant of policy iteration algorithm. To start our proof for Theorem 1, we first derive the following lemma,

**Lemma 1.** *If $\zeta = \max_s |\bar{A}^{\pi, \tilde{\pi}}(s)|$, then*

$$\left\| J(\pi) - \tilde{J}(\pi, \tilde{\pi}) \right\|_1 \le 2\zeta \frac{\gamma}{(1-\gamma)^2} D_{TV}^{\max}(\tilde{\pi}, \pi) \le 2\zeta \frac{\gamma}{(1-\gamma)^2} \sqrt{D_{KL}^{\max}(\tilde{\pi}, \pi)}$$

*Proof.* We define $\rho_t^\pi(s_t)$ as the marginal state distribution at time $t$ assuming that the agent follows policy $\pi$ from initial state $\rho_0(s_t)$ at time $t = 0$. Note that from the definition of $\rho^\pi$, $\rho^\pi(s) = \sum_{t=0}^{\infty} \gamma^t \rho_t^\pi(s_t = s)$. We can use the following lemma from Kahn et al. (2016), which is adapted from Ross et al. (2011) and Schulman et al. (2015).

**Lemma 2.** *(Kahn et al., 2016)*

$$\left\| \rho_t^\pi - \rho_t^\beta \right\|_1 \le 2t D_{TV}^{\max}(\pi, \beta) \le 2t \sqrt{D_{KL}^{\max}(\pi, \beta)} \tag{1}$$

Using Lemma 2, the full proof for Lemma 1 is provided below, where $\bar{A}^{\pi,\tilde{\pi}}(s) = \mathbb{E}_\pi[A^{\tilde{\pi}}(s_t, a_t)]$ and $A^{\tilde{\pi}}(s_t, a_t)$ is the advantage function of $\tilde{\pi}$,

$$
\begin{aligned}
&\left\| J(\pi) - \tilde{J}(\pi, \tilde{\pi}) \right\|_1 \\
&= \left\| \mathbb{E}_{\rho^{\tilde{\pi}}}[\bar{A}^{\pi,\tilde{\pi}}(s)] - \mathbb{E}_{\rho^\pi}[\bar{A}^{\pi,\tilde{\pi}}(s)] \right\|_1 \\
&\leq \sum_{t=0}^{\infty} \gamma^t \left\| \mathbb{E}_{\rho_t^{\tilde{\pi}}}[\bar{A}^{\pi,\tilde{\pi}}(s)] - \mathbb{E}_{\rho_t^\pi}[\bar{A}^{\pi,\tilde{\pi}}(s)] \right\|_1 \\
&\leq \zeta \sum_{t=0}^{\infty} \gamma^t \left\| \rho_t^{\tilde{\pi}} - \rho_t^\pi \right\|_1 \\
&\leq 2\zeta (\sum_{t=0}^{\infty} \gamma^t t) D_{TV}^{\max}(\tilde{\pi}, \pi) \\
&= 2\zeta \frac{\gamma}{(1-\gamma)^2} D_{TV}^{\max}(\tilde{\pi}, \pi) \\
&\leq 2\zeta \frac{\gamma}{(1-\gamma)^2} \sqrt{D_{KL}^{\max}(\pi, \tilde{\pi})}. \quad \square
\end{aligned}
\tag{2}
$$

This lemma is crucial in our theoretical analysis, as it allows us to tractably bound the biases of the full spectrum of local IPG objectives $\tilde{J}^{\beta,\nu,CV}(\pi, \tilde{\pi})$ against $J(\pi)$.

## A.2  Main proof for Theorem 1

*Proof.* We first prove the bound for $\left\| \tilde{J}(\pi, \tilde{\pi}) - \tilde{J}^{\beta,\nu=0,CV}(\pi, \tilde{\pi}) \right\|_1$. Using Lemma 2, the bound is given below, with a similar derivation process as in Lemma 1.

$$
\begin{aligned}
&\left\| \tilde{J}(\pi, \tilde{\pi}) - \tilde{J}^{\beta,\nu=0,CV}(\pi, \tilde{\pi}) \right\|_1 \\
&= \left\| J(\tilde{\pi}) + \mathbb{E}_{\rho^{\tilde{\pi}},\pi}[A^{\tilde{\pi}}(s_t, a_t)] - J(\tilde{\pi}) - \mathbb{E}_{\rho^{\tilde{\pi}},\pi}[A^{\tilde{\pi}}(s_t, a_t) - A_w^{\tilde{\pi}}(s_t, a_t)] - \mathbb{E}_{\rho^\beta}[\bar{A}_w^{\pi,\tilde{\pi}}(s_t)] \right\|_1 \\
&= \left\| \mathbb{E}_{\rho^{\tilde{\pi}}}[\bar{A}_w^{\pi,\tilde{\pi}}(s)] - \mathbb{E}_{\rho^\beta}[\bar{A}_w^{\pi,\tilde{\pi}}(s)] \right\|_1 \\
&\leq \sum_{t=0}^{\infty} \gamma^t \left\| \mathbb{E}_{\rho_t^{\tilde{\pi}}}[\bar{A}_w^{\pi,\tilde{\pi}}(s)] - \mathbb{E}_{\rho_t^\beta}[\bar{A}_w^{\pi,\tilde{\pi}}(s)] \right\|_1 \\
&\leq \epsilon \sum_{t=0}^{\infty} \gamma^t \left\| \rho_t^{\tilde{\pi}} - \rho_t^\beta \right\|_1 \\
&\leq 2\epsilon (\sum_{t=0}^{\infty} \gamma^t t) D_{TV}^{\max}(\tilde{\pi}, \beta) \\
&= 2\epsilon \frac{\gamma}{(1-\gamma)^2} D_{TV}^{\max}(\tilde{\pi}, \beta) \\
&\leq 2\epsilon \frac{\gamma}{(1-\gamma)^2} \sqrt{D_{KL}^{\max}(\tilde{\pi}, \beta)}.
\end{aligned}
\tag{3}
$$

Given this bound, we can directly derive the bound for $\left\| \tilde{J}(\pi, \tilde{\pi}) - \tilde{J}^{\beta,\nu=0,CV}(\pi, \tilde{\pi}) \right\|_1$ by combining with Lemma 1,

$$
\begin{aligned}
&\left\| J(\pi) - \tilde{J}^{\beta,\nu=0,CV}(\pi, \tilde{\pi}) \right\|_1 \\
&\left\| J(\pi) - \tilde{J}(\pi, \tilde{\pi}) + \tilde{J}(\pi, \tilde{\pi}) - \tilde{J}^{\beta,\nu=0,CV}(\pi, \tilde{\pi}) \right\|_1 \\
&\leq \left\| \tilde{J}(\pi, \tilde{\pi}) - \tilde{J}^{\beta,\nu=0,CV}(\pi, \tilde{\pi}) \right\|_1 + \left\| J(\pi) - \tilde{J}(\pi, \tilde{\pi}) \right\|_1 \\
&\leq 2 \frac{\gamma}{(1-\gamma)^2} \left( \epsilon \sqrt{D_{KL}^{\max}(\tilde{\pi}, \beta)} + \zeta \sqrt{D_{KL}^{\max}(\pi, \tilde{\pi})} \right) \quad \square
\end{aligned}
\tag{4}
$$

# B  Algorithm with Monotonic Convergence Property and its Proof

---

**Algorithm 1** Policy iteration with non-decreasing returns $J(\pi)$ and bounded off-policy sampling

---

1: Initialize policy $\pi_0$, and critic $Q_w$
2: **repeat**
3:     Compute all advantage values $A^{\pi_i}(s, a)$, and choose any off-policy distribution $\beta_i$
4:     Update critic $Q_w$ using any method (no requirement for performance)
5:     Solve the constrained optimization problem:
6:         $\pi_{i+1} \leftarrow \arg\max_\pi \tilde{J}^{\beta_i, \nu=0, CV}(\pi, \pi_i) - C\left( \zeta\sqrt{D_{KL}^{\max}(\pi, \pi_i)} + \epsilon\sqrt{D_{KL}^{\max}(\pi_i, \beta_i)} \right)$
7:         subject to $\sum_a \pi(a|s) = 1 \quad \forall s$
8:         where $C = \frac{2\gamma}{(1-\gamma)^2}, \zeta = \max_s |\bar{A}^{\pi, \tilde{\pi}}(s)|, \epsilon = \max_s |\bar{A}_w^{\pi, \tilde{\pi}}(s)|$
9: **until** $\pi_i$ converges.

---

Algorithm 1 is a special case of IPG, $\tilde{J}^{\beta_i, \nu=0, CV}(\pi, \pi_i)$. We can prove that Algorithm 1 guarantees monotonic improvement, even with off-policy sample usage and imperfect critic $Q_w$ or $A_w$. This is an interesting result, since most of the prior work have shown such property only for purely on-policy policy gradient methods Kakade & Langford (2002); Schulman et al. (2015). We begin by first introducing the following corollary,

**Corollary 1.**

$$J(\pi) \geq M(\pi, \tilde{\pi}) \geq M^{\beta, \nu=0, CV}(\pi, \tilde{\pi}), J(\tilde{\pi}) = M(\tilde{\pi}, \tilde{\pi}) = M^{\beta, \nu=0, CV}(\tilde{\pi}, \tilde{\pi}) \tag{5}$$

*where*

$$M(\pi, \tilde{\pi}) = \tilde{J}(\pi, \tilde{\pi}) - C\zeta\sqrt{D_{KL}^{\max}(\pi, \tilde{\pi})}$$

$$M^{\beta, \nu=0, CV}(\pi, \tilde{\pi}) = \tilde{J}^{\beta, \nu=0}(\pi, \tilde{\pi}) - C(\zeta\sqrt{D_{KL}^{\max}(\pi, \tilde{\pi})} + \epsilon\sqrt{D_{KL}^{\max}(\tilde{\pi}, \beta)})$$

$$C = \frac{2\gamma}{(1-\gamma)^2}, \zeta = \max_s |\bar{A}^{\pi, \tilde{\pi}}(s)|, \epsilon = \max_s |\bar{A}_w^{\pi, \tilde{\pi}}(s)|$$

*Proof.* It follows from Theorem 1 in the main text and Theorem 1 in Schulman et al. (2015). $J(\tilde{\pi}) = M^{\beta, \nu=0, CV}(\tilde{\pi}, \tilde{\pi})$ since $\zeta = \epsilon = 0$ when $\pi = \tilde{\pi}$.

Given Corollary 1, we use minorization-maximization (MM) (Hunter & Lange, 2004) to derive Algorithm 2, a policy iteration algorithm that allows using off-policy samples while guaranteeing monotonic improvement on $J(\pi)$. MM suggests that at each iteration, by maximizing the lower bound, or the minorizer, of the objective, the algorithm can guarantee monotonic improvement: $J(\pi_{i+1}) \geq M^{\beta_i, \nu=0, CV}(\pi_{i+1}, \pi_i) \geq M^{\beta_i, \nu=0, CV}(\pi_i, \pi_i) = J(\pi_i)$, where $\pi_{i+1} \leftarrow \arg\max_\pi M^{\beta_i, \nu=0, CV}(\pi, \pi_i)$. Importantly, the algorithm guarantees monotonic improvement regardless of the off-policy distribution $\beta_i$ or the performance of the critic $Q_w$. This result is a step toward achieving off-policy policy gradient with convergence guarantee of on-policy algorithms.[1]

We compare our theoretical algorithm with Algorithm 1 in Schulman et al. (2015), which guarantees monotonic improvement in a general on-policy policy gradient algorithm. The main difference is the additional term, $-C\epsilon\sqrt{D_{KL}^{\max}(\tilde{\pi}, \beta)}$ to the lower bound. $D_{KL}^{\max}(\tilde{\pi}, \beta)$ is constant with respect to $\pi$, while $\epsilon = 0$ if $\pi = \tilde{\pi}$ and $\epsilon \geq 0$ if otherwise. This suggests that as $\beta$ becomes more off-policy, the gap between the lower bound and the true objective widens, proportionally to $\sqrt{D_{KL}^{\max}(\tilde{\pi}, \beta)}$. This may make each majorization step end in a place very close to where it started, i.e. $\pi_{i+1}$ very close to $\pi_i$, and slow down learning. This again suggests a trade-off that comes in as off-policy samples are used.

# C  Proof for Theorem 2

We follow the same procedure as the proof for Theorem 1, where we first derive bounds between $\tilde{J}(\pi, \tilde{\pi})$ and the other local objectives, and then combine the results with Lemma 1.

To begin the proof, we first derive the bound for the special case where $\nu = 1$. Having $\nu = 1$, we remove the likelihood ratio policy gradient term, and get the following gradient expression,

$$\nabla_\theta J(\theta) \approx \mathbb{E}_{\rho^\beta}[\nabla_\theta \bar{Q}_w^\pi(s_t)]. \tag{6}$$

This is an off-policy actor-critic algorithm, and is closely connected to DDPG (Lillicrap et al., 2016), except that it does not use target policy network and its use of a stochastic policy enables on-policy exploration, trust-region policy updates, and no heuristic additive exploration noise.

We can introduce the following bound on the local objective $\tilde{J}^{\beta,\nu=1}(\pi, \tilde{\pi})$, whose policy gradient equals 6 at $\pi = \tilde{\pi}$, similarly to the proof for Theorem 1 in the main text.

**Corollary 2.** *If* $\delta = \max_{s,a} |A^{\tilde{\pi}}(s,a) - A_w^{\tilde{\pi}}(s,a)|$, $\epsilon = \max_s |\bar{A}_w^{\pi,\tilde{\pi}}(s)|$, *and*

$$\tilde{J}^{\beta,\nu=1}(\pi, \tilde{\pi}) = J(\tilde{\pi}) + \mathbb{E}_{\rho^\beta}[\bar{A}_w^{\pi,\tilde{\pi}}(s_t)] \tag{7}$$

*then,*

$$\left\| \tilde{J}(\pi, \tilde{\pi}) - \tilde{J}^{\beta,\nu=1}(\pi, \tilde{\pi}) \right\|_1 \leq \frac{\delta}{1-\gamma} + 2\epsilon \frac{\gamma}{(1-\gamma)^2} \sqrt{D_{KL}^{\max}(\tilde{\pi}, \beta)} \tag{8}$$

*Proof.* We note that

$$
\begin{aligned}
&\left\| \tilde{J}(\pi, \tilde{\pi}) - \tilde{J}^{\beta,\nu=1}(\pi, \tilde{\pi}) \right\|_1 \\
&= \left\| \mathbb{E}_{\rho^{\tilde{\pi}},\pi}[A^{\tilde{\pi}}(s_t, a_t) - A_w^{\tilde{\pi}}(s_t, a_t)] + \tilde{J}(\pi, \tilde{\pi}) - \tilde{J}^{\beta,\nu=0}(\pi, \tilde{\pi}) \right\|_1 \\
&\leq \left\| \mathbb{E}_{\rho^{\tilde{\pi}},\pi}[A^{\tilde{\pi}}(s_t, a_t) - A_w^{\tilde{\pi}}(s_t, a_t)] \right\|_1 + \left\| \tilde{J}(\pi, \tilde{\pi}) - \tilde{J}^{\beta,\nu=0}(\pi, \tilde{\pi}) \right\|_1 \\
&\leq \sum_{t=0}^{\infty} \gamma^t \left\| \mathbb{E}_{\rho_t^{\tilde{\pi}},\pi}[A^{\tilde{\pi}}(s_t, a_t) - A_w^{\tilde{\pi}}(s_t, a_t)] \right\|_1 + \left\| \tilde{J}(\pi, \tilde{\pi}) - \tilde{J}^{\beta,\nu=0}(\pi, \tilde{\pi}) \right\|_1 \\
&\leq \delta \sum_{t=0}^{\infty} \gamma^t + \left\| \tilde{J}(\pi, \tilde{\pi}) - \tilde{J}^{\beta,\nu=0}(\pi, \tilde{\pi}) \right\|_1 \\
&= \frac{\delta}{1-\gamma} + \left\| \tilde{J}(\pi, \tilde{\pi}) - \tilde{J}^{\beta,\nu=0}(\pi, \tilde{\pi}) \right\|_1 \\
&\leq \frac{\delta}{1-\gamma} + 2\epsilon \frac{\gamma}{(1-\gamma)^2} \sqrt{D_{\text{KL}}^{\max}(\tilde{\pi}, \beta)}, \quad \square
\end{aligned}
\tag{9}
$$

where the proof uses Theorem 1 at the last step.

Given Corollary 2 and Theorem 1, we are ready to prove the two bounds in Theorem 2.

*Proof.*

$$\left\|\tilde{J}(\pi, \tilde{\pi}) - \tilde{J}^{\beta,\nu}(\pi, \tilde{\pi})\right\|_1$$

$$= \left\|J(\tilde{\pi}) + \mathbb{E}_{\rho^{\tilde{\pi}}, \pi}[A^{\tilde{\pi}}(s_t, a_t)] - J(\tilde{\pi}) - (1-\nu)\mathbb{E}_{\rho^{\tilde{\pi}}, \pi}[A^{\tilde{\pi}}(s_t, a_t)] - \nu\mathbb{E}_{\rho^{\beta}}[\bar{A}_w^{\pi,\tilde{\pi}}(s_t)]\right\|_1$$

$$= \nu \left\|\mathbb{E}_{\rho^{\tilde{\pi}}, \pi}[A^{\tilde{\pi}}(s_t, a_t)] - \mathbb{E}_{\rho^{\beta}}[\bar{A}_w^{\pi,\tilde{\pi}}(s_t)]\right\|_1$$

$$= \nu \left\|\mathbb{E}_{\rho^{\tilde{\pi}}, \pi}[A^{\tilde{\pi}}(s_t, a_t)] - \mathbb{E}_{\rho^{\pi}}[\bar{A}_w^{\pi,\tilde{\pi}}(s_t)] + \mathbb{E}_{\rho^{\pi}}[\bar{A}_w^{\pi,\tilde{\pi}}(s_t)] - \mathbb{E}_{\rho^{\beta}}[\bar{A}_w^{\pi,\tilde{\pi}}(s_t)]\right\|_1$$

$$\leq \nu \left\|\mathbb{E}_{\rho^{\tilde{\pi}}, \pi}[A^{\tilde{\pi}}(s_t, a_t)] - \mathbb{E}_{\rho^{\pi}}[\bar{A}_w^{\pi,\tilde{\pi}}(s_t)]\right\|_1 + \nu \left\|\mathbb{E}_{\rho^{\pi}}[\bar{A}_w^{\pi,\tilde{\pi}}(s_t)] - \mathbb{E}_{\rho^{\beta}}[\bar{A}_w^{\pi,\tilde{\pi}}(s_t)]\right\|_1$$

$$= \nu \left\|\mathbb{E}_{\rho^{\tilde{\pi}}, \pi}[A^{\tilde{\pi}}(s_t, a_t) - \bar{A}_w^{\tilde{\pi}}(s_t, a_t)]\right\|_1 + \nu \left\|\mathbb{E}_{\rho^{\pi}}[\bar{A}_w^{\pi,\tilde{\pi}}(s_t)] - \mathbb{E}_{\rho^{\beta}}[\bar{A}_w^{\pi,\tilde{\pi}}(s_t)]\right\|_1$$

$$\leq \frac{\nu\delta}{1-\gamma} + 2\epsilon\frac{\nu\gamma}{(1-\gamma)^2}\sqrt{D_{\mathrm{KL}}^{\max}(\tilde{\pi}, \beta)}$$

$$\left\|\tilde{J}(\pi, \tilde{\pi}) - \tilde{J}^{\beta,\nu,CV}(\pi, \tilde{\pi})\right\|_1$$

$$= \left\|J(\tilde{\pi}) + \mathbb{E}_{\rho^{\tilde{\pi}}, \pi}[A^{\tilde{\pi}}(s_t, a_t)] - J(\tilde{\pi}) - (1-\nu)\mathbb{E}_{\rho^{\tilde{\pi}}, \pi}[A^{\tilde{\pi}}(s_t, a_t) - \bar{A}_w^{\tilde{\pi}}(s_t, a_t)] - \mathbb{E}_{\rho^{\beta}}[\bar{A}_w^{\pi,\tilde{\pi}}(s_t)]\right\|_1$$

$$= \left\|\nu(\mathbb{E}_{\rho^{\tilde{\pi}}, \pi}[A^{\tilde{\pi}}(s_t, a_t)] - \mathbb{E}_{\rho^{\pi}}[\bar{A}_w^{\pi,\tilde{\pi}}(s_t)]) + \mathbb{E}_{\rho^{\pi}}[\bar{A}_w^{\pi,\tilde{\pi}}(s_t)] - \mathbb{E}_{\rho^{\beta}}[\bar{A}_w^{\pi,\tilde{\pi}}(s_t)]\right\|_1$$

$$\leq \nu \left\|\mathbb{E}_{\rho^{\tilde{\pi}}, \pi}[A^{\tilde{\pi}}(s_t, a_t)] - \mathbb{E}_{\rho^{\pi}}[\bar{A}_w^{\pi,\tilde{\pi}}(s_t)]\right\|_1 + \left\|\mathbb{E}_{\rho^{\pi}}[\bar{A}_w^{\pi,\tilde{\pi}}(s_t)] - \mathbb{E}_{\rho^{\beta}}[\bar{A}_w^{\pi,\tilde{\pi}}(s_t)]\right\|_1$$

$$\leq \frac{\nu\delta}{1-\gamma} + 2\epsilon\frac{\gamma}{(1-\gamma)^2}\sqrt{D_{\mathrm{KL}}^{\max}(\tilde{\pi}, \beta)}. \tag{10}$$

We combine these bounds with Lemma 1 to conclude the proof.

## D  Control Variates for Policy Gradient

In this Section, we describe control variate choices for policy gradient methods other than the linear case presented in Q-Prop (Gu et al., 2017).

### D.1  Reparameterized Critic Control Variate

If the action is continuous and the policy is a simple distribution such as a Gaussian, one option is to use the full $Q_w$ as the control variate and use Monte Carlo to estimate its expectation with respect to the policy. A significant reduction in the variance could still be possible by applying the reparameterization trick (Kingma & Welling, 2014) on the correction term. For example, given a univariate Gaussian policy $\pi_\theta(a_t|s_t) = \mathcal{N}(\mu_\theta(s_t), \sigma_\theta(s_t))$,

$$\bar{Q}_w^\pi(s_t) = \mathbb{E}_\pi[Q_w(s_t, a_t)] = \mathbb{E}_{\epsilon \sim \mathcal{N}(0,1)}[Q_w(s_t, \mu_\theta(s_t) + \epsilon\sigma_\theta(s_t))]$$

$$\approx \frac{1}{m}\sum_{i=1}^m Q_w(s_t, \mu_\theta(s_t) + \epsilon_i\sigma_\theta(s_t)). \tag{11}$$

This can be applied for multivariate Gaussian policies and any other reparametrizable action distributions (Kingma & Welling, 2014).

### D.2  Discrete Critic Control Variate

For a simple categorical action distribution, computing the expectation of the Q-function under the stochastic policy is straight-forward. Let $\boldsymbol{\pi}_\theta(s_t) \in \mathbb{R}^k$ denote a probability vector over $k$ discrete actions, and $\boldsymbol{Q}_w(s_t) \in \mathbb{R}^k$ denote the action-value function for the $k$ actions, as in DQN (Mnih et al., 2015), the correction term to be differentiated is,

$$\bar{Q}_w^\pi(s_t) = \boldsymbol{\pi}_\theta(s_t)^T \cdot \boldsymbol{Q}_w(s_t). \tag{12}$$

### D.3  NAF Critic Control Variate

Continuous control problems often parametrize the policy as a multivariate Gaussian. In such case, we may propose a critic that is quadratic with respect to the action, and get analytic expectation.

For example, given $\pi_\theta(a_t|s_t) = \mathcal{N}(\mu_\theta(s_t), \Sigma_\theta(s_t))$, the quadratic $Q_w$ from Normalized Advantage Function (NAF) (Gu et al., 2016) can be used as the Q-function parametrization,

$$Q_w(s_t, a_t) = A_w(s_t, a_t) + V_w(s_t)$$
$$A_w(s_t, a_t) = -\frac{1}{2}(a_t - \mu_w(s_t))^T P_w(s_t)(a_t - \mu_w(s_t)). \tag{13}$$

The correction term can then be computed analytically, without sampling required as in reparametrized critic control variates in Section D.1.

$$\bar{Q}_w^\pi(s_t) = V_w(s_t) - \frac{1}{2}\text{Tr}(P_w(s_t)\Sigma_\theta(s_t))$$
$$- \frac{1}{2}(\mu_\theta(s_t) - \mu_w(s_t))^T P_w(s_t)(\mu_\theta(s_t) - \mu_w(s_t)). \tag{14}$$

# E Supplementary Experimental Details

## E.1 Hyperparameters

GAE($\lambda = 0.97$) (Schulman et al., 2016) is used for $\hat{A}$ estimation. Trust-region update in TRPO is used as the policy optimizer (Schulman et al., 2015). The standard Q-fitting routine from DDPG (Lillicrap et al., 2016) is used for fitting $Q_w$, where $Q_w$ is trained with batch size 64, using experience replay of size $1e6$, and target network with $\tau = 0.001$. ADAM (Kingma & Ba, 2014) is used as the optimizer for $Q_w$. Policy network parametrizes a Gaussian policy with $\pi_\theta(a_t|s_t) = \mathcal{N}(\mu_\theta(s_t), \Sigma_\theta)$, where $\mu_\theta$ is a two-hidden-layer neural network of size $100 - 50$ and tanh hidden nonlinearity and linear output, and $\Sigma_\theta$ is a diagonal, state-independent variance. For DDPG, the policy network is deterministic and additionally has tanh activation at the output layer. The critic function $Q_w$ is a two-hidden-layer neural network of size $100 - 100$ with ReLU activation. For IPG methods with the control variates, we further explored the standard and conservative variants, the technique proposed in Q-Prop (Gu et al., 2017), and the Taylor expansion variant with the reparametrized variant discussed in Section D.1. For the reparameterized control variates, we use Monte Carlo sample size $m = 1$.

The trust-region step size for policy update is fixed to $0.1$ for HalfCheetah-v1 and Humanoid-v1, and $0.01$ for Ant-v1 and Walker2d-v1, while the learning rate for ADAM in critic update is fixed to $1e^{-4}$ for HalfCheetah-v1, Ant-v1, Humanoid-v1, and $1e^{-3}$ for Walker2d-v1. Those two hyperparameters are found by first running TRPO and DDPG on each domain, and picking the ones that give best performance for each domain. These parameters are fixed throughout the experiment to ensure fair comparisons.

As in the Q-Prop implementation (Gu et al., 2017), the residual learning signal in the first term is normalized to be zero mean and unit variance. This introduces additional bias to the gradient estimator, but the bias can be theoretically analyzed by substituting the bounds with new $\nu' = 1 - \frac{1-\nu}{\sigma}$ in the IPG expressions where $\sigma$ is the empirical standard deviation of the (residual) learning signal.

The plots in the main text present the mean returns as solid lines, scatter plots of all runs in the background to visualize variability. For $X$-axis, one "episode" corresponds to 1000 transitions, which is the default maximum episode length for all domains in our experiments. Importantly, "Episodes" do not correspond to actual numbers of episodes taken for Ant-v1, Walker-v1, and Humanoid-v1, since these environments have termination conditions.

## E.2 Additional Plot

Figure 1 shows additional plot on Humanoid-v1.

Figure 1: IPG-$\nu = 0.2$-$\pi$-CV vs Q-Prop and TRPO on Humanoid-v1 with batch size 10000 in the first 10000 episodes. IPG-$\nu = 0.2$-$\pi$-CV, with a small difference of $\nu = 0.2$ multiplier, out-performs Q-Prop. All these methods have stable, monotonic policy improvement. The experiment is cut at 10000 episodes due to heavy compute requirement of Q-Prop and IPG methods, mostly from fitting the off-policy critic.

## Footnotes

[1]Schulman et al. (2015) applies additional bound, $\epsilon \geq 2\epsilon'\sqrt{D_{KL}^{\max}(\pi, \tilde{\pi})}$ where $\epsilon' = \max_{s,a} |A_w^{\tilde{\pi}}(s, a)|$ to remove dependency on $\pi$. In our case, we cannot apply such bound on $\zeta$, since then the inequality in Theorem 1 is still satisfied but the equality is violated, and thus the algorithm no longer guarantees monotonic improvement.