[Reviews · NeurIPS 2017]

Reviewer 1



* summary: This paper studies how to merge on-policy and off-policy gradient algorithms. The authors propose a new policy gradient framework that unifies many previous on-policy and off-policy gradient methods. Many previous policy gradient algorithms can not only be re-derived and but also get improved by the introduction of control variate. Even though this framework introduces bias to gradient updates but, they show in theoretical results that this bias can be bounded. Experiments are very well done and provide enough insights to understand the proposed framework. * In overall, the paper is very well written with excellent related work and discussions. The main contents are well presented as well, the theoretical results are very intuitive and make sense. * I think this is very strong paper. I have only few comments as following: - I think that even if one is familiar with TRPO, he might still not be able to know how to transfer the technique of TRPO to approximately implement Algorithm 2. It would be nicer if the authors elaborate on this step. - As seen in Eq. (6) and Theorem 1, the bias introduced by the difference between \beta and \pi is non-trivial. Therefore, I find that IPG does not really always outperform Q-Prop, sometimes they are comparable (when \beta is not equal to \pi). According to my understanding, v from 0 to 1 means a shifting gradually from on-policy to off-policy, therefore one could compensate the bias by gathering more data/larger batch-size. Therefore, it might be more interesting to see if Table 2 is expanded with more experiment setting: v increases gradually from 0 to 1 (with and without control variate), and also with the change of data-size/batch-size to see if bias can be compensated somehow. - line 151, should it be $v=0$?

Reviewer 2



As the title says, this paper proposes a framework to merge on-policy and off-policy gradient estimation for deep reinforcement learning to achieve better sample efficiency and stability. The topic addressed in the paper is of importance, the paper is overall well-written, the experiments done in the paper looks fair. Experimental results show that a reasonable setting for IPG tends to depends on tasks. Since it is always annoying to tune these setting, it would be nice to provide a default setting and provide when and how to modify these parameters.

Reviewer 3



I have read the author's rebuttal and chosen not to change any of my review below. This paper presents a method for using both on and off-policy data when estimating the policy gradient. The method, called Interpolated Policy Gradient, is presented in Equation (5). In this equation the first term is the standard on-policy policy gradient. The second term is a control variate that can use samples from various different distributions (e.g., on or off policy). Different definitions of \beta, which denotes the policy that s_t is sampled according to, causes this algorithm to become equivalent to various previous methods. The paper is well written, presents results that are novel, and clearly places the new method in the context of related literature. The empirical results support the theoretical claims. I recommend acceptance. One thing that would have helped me a lot would be if there was more clarity in the use of $\pi$ and $\theta$. For example, in equation (5), I think of having my current policy parameters, $\theta$, and some other old policy parameters ,$\theta_old$. Then I read the first term and see E_{\rho^\pi}. This term really means $E_{\rho^{\pi_\theta}}$, since $\pi_\theta$ is the policy, not $\pi$. Dropping the \theta also hides the important term here - this first term uses samples from $\pi_\theta$, not $\pi_{\theta_old}$. That is, $\pi$ isn't the important term here, $\theta$ is, and $\theta$ isn't in the subscript on the expected value. I suggest using $E_{\rho^{\pi_\theta}}$ or $E_{\rho^\theta}$, and saying that $\rho^\theta$ is shorthand for $\rho^{\pi_\theta}$. On line 107 "IHowever" should be "However". The green text in Algorithm 1 is hard to see.